# Fast and Efficient Method for Optical Coherence Tomography Images Classification Using Deep Learning Approach

**DOI:** 10.3390/s22134675

**Published:** 2022-06-21

**Authors:** Rouhollah Kian Ara, Andrzej Matiolański, Andrzej Dziech, Remigiusz Baran, Paweł Domin, Adam Wieczorkiewicz

**Affiliations:** 1Institute of Telecommunications, AGH University of Science and Technology, 30-059 Krakow, Poland; kian.agh20@gmail.com (R.K.A.); andrzej.dziech@agh.edu.pl (A.D.); 2Faculty of Electrical Engineering, Automatic Control and Computer Science, Kielce University of Technology, 25-314 Kielce, Poland; r.baran@tu.kielce.pl; 3Consultronix S.A., 32-083 Balice, Poland; pdomin@cxsa.pl (P.D.); awieczorkiewicz@cxsa.pl (A.W.)

**Keywords:** artificial neural networks, biomedical imaging, image analysis, optical coherence tomography, oct, convolutional neural network

## Abstract

The use of optical coherence tomography (OCT) in medical diagnostics is now common. The growing amount of data leads us to propose an automated support system for medical staff. The key part of the system is a classification algorithm developed with modern machine learning techniques. The main contribution is to present a new approach for the classification of eye diseases using the convolutional neural network model. The research concerns the classification of patients on the basis of OCT B-scans into one of four categories: Diabetic Macular Edema (DME), Choroidal Neovascularization (CNV), Drusen, and Normal. Those categories are available in a publicly available dataset of above 84,000 images utilized for the research. After several tested architectures, our 5-layer neural network gives us a promising result. We compared them to the other available solutions which proves the high quality of our algorithm. Equally important for the application of the algorithm is the computational time, which is reduced by the limited size of the model. In addition, the article presents a detailed method of image data augmentation and its impact on the classification results. The results of the experiments were also presented for several derived models of convolutional network architectures that were tested during the research. Improving processes in medical treatment is important. The algorithm cannot replace a doctor but, for example, can be a valuable tool for speeding up the process of diagnosis during screening tests.

## 1. Introduction

The current development of methods based on artificial intelligence enables the creation of new data analysis solutions. One example of the use of these methods is the automation of medical diagnostics. This trend is apparent from the heavy investment in application solutions for image analysis by companies such as Google [1] and GE Healthcare [2]. Many of them, such as facial recognition, plate detection, and people counting, have already been commercialized. Nevertheless, there is still room for improving the operation of the algorithms to obtain the most effective solutions. One of the areas in which the effectiveness and reliability of analysis are of the greatest importance is health protection.

The topicality and the huge potential benefits of using artificial intelligence methods in medicine were confirmed by many publications. García and Simunic [3] show great potential for automatic diagnostics, subject to the certainty of diagnosis. Pesapane et al. [4] present an analysis of the chances for the development of systems supporting medical personnel in the near future by reducing diagnostic cost and time.

Optical Coherence Tomography (OCT) is an imaging technique that uses low-coherence light to capture two and three-dimensional images in micrometer-resolution. OCT is a widely used medical imaging method for capturing retina images. There are three types of OCT scans. Firstly A-scans (1D) are merged to produce B-scans with 2D information about the surface. Thirdly C-scans are 3D representations of B-Scans.

OCT is a non-invasive imaging test. It shows each of the characteristic layers of the retina. This allows to map and measure its thickness and recognize pathologies. OCT is very useful in the diagnosis of diseases, such as Diabetic Macular Edema (DME), Choroidal NeoVascularization (CNV), and Drusen [5,6].

In summary, this paper has the following contributions:In this paper, we propose an artificial intelligence aid solution for medical image classification. We deal with multiple classifications among four classes of images: DME, CNV, Drusen, and Normal (without visible pathology). Figure 1 presents images from the dataset used in the research taken from a publicly available dataset [7].We present an optimized implementation of the CNN model for medical image (OCT) analysis. We conduct experiments on public datasets [7]. Experimental results show that the proposed approach achieves high accuracy compared to the state-of-the-art algorithms.It is a novel study that emphasizes the importance of using augmented data in the training of the OCT images rather than increasing the depth (number of hidden layers) and width (number of filters) of the model.

The rest of this paper is organized as follows. Section 2 briefly reviews the relevant studies. Section 3 explains the database and its differences from the augmented dataset. Section 4 describes the proposed models. Section 5 evaluates our experiments and results. Section 6 discusses the significance of the proposed approach. Lastly, Section 7 concludes the paper.

## 2. Related Works

Medical image processing is an important field of research. Many researchers deal with the problem related to its processing such as the acquisition of artifacts, segmentation, or feature extraction. Sánchez et al. [8] deal with motion artifacts in OCT imaging. Phadikar et al. [9] proposed a solution for muscle artifacts removal. Interesting solutions for segmentation were proposed by Ahmad et al. [10] and Qadri et al. [11]. Ahmed et al. [12] and Zhang et al. [13] introduce novel methods for automatic features extraction.

Machine learning approaches to analyze OCT images are already present in the scientific literature [14]. For example, Schmidt-Erfurth et al. [15] discuss the comparison of unsupervised and supervised learning in the classification of patients with macular degeneration (AMD) and diabetic retinopathy (DR). The authors demonstrate the great possibilities in the application of deep learning. An overview of the available AMD classification solutions introduced until 2018 was presented by Ting et. al. [16]. Five solutions are described, the best of which offered 97% efficiency for binary classification (healthy vs. AMD) over approximately 20 thousand images in the database. Lee et al. [17] described a 21-layer convolutional neural network (CNN) for grading AMD disease. In the binary classification (AMD vs. healthy) they obtained 93% of accuracy. Kermany et al. [7] proposed a CNN solution based on the Inception V3 model using transfer learning reporting 96.6% accuracy. The database contains approximately 84 thousand samples divided into Normal, Drusen, CNV, and DME categories. An unsupervised learning approach was described by Seeböck et al. [18], using denoised images and one-class SVM. The method results in a noticeably lower accuracy—81.4%. Classification based on a layer-guided convolutional neural network to classify Normal retina, CNV, DME, or Drusen was proposed by Huang et al. [19]. The solution includes a network for segmentation and classification giving a final result at the 89.9% level. Different approaches using fuzzy c-means segmentation were presented by Chowdhary and Acharja [20,21]. Das et al. [22] proposed a CNN model for the classification of AMD and DME using multi-scale deep features fusion (MDFF). The final accuracy of 99.6% was achieved on an approximately 84-thousand-sample database. An approach based on the Inception v3 model was proposed by Hwang et al. [23]. The CNN model was trained with pre-processed images resulting in a 96.9%. Tasnim et al. [24] present a study of a deep learning approach to retinal OCT images. They described several models including MobileNetV2 with 99.17% accuracy with the  Kermany database [7] on over 80k samples divided into 4 categories. Kaymak and Serener [25] adopted the AlexNet architecture for automatic classification of categories of diseases (DME, CNV, Drusen) and healthy patients. The final accuracy obtained was 97.7%. A study by Li et al. [26] on the same database using multi ResNet50 architecture provides at best 97.9% accuracy. The authors perform a 10-fold cross-validation and proposed visualization based on saliency maps. Lo et al. [27] aim to identify only the epiretinal membrane (ERM) in OCT images. The authors described a solution based on ResNet-101 resulting in 98% accuracy over approximately 4.5k samples divided into two categories. Tsuji et al. [28] show promising results for OCT image classification, using the capsule network and InceptionV3 achieving 99.6% and 99.8% accuracy, respectively. The OctNET model proposed by Prabhakaran et al. [29] introduced a new architecture. The authors performed research on the Kermany database and achieved good performance on the data—99.69%. Their model is also relatively light, with the potential for quick computations.

Considering related works our study aims to propose a robust solution for disease classification based on OCT images. We are aware that an applicable solution must be highly effective and fast in computations. Therefore, we are proposing in the paper a high-accuracy solution with a very light CNN model. It implicates our primary goal to achieve top classification effectiveness with limited computational resources. To enable our research to be readily compared and make it reproducible, our solution is presented utilizing a publicly available database.

## 3. Database and Augmentation

The database used for the current research is available to download as described in [7]. It contains raw OCT images stored in JPEG format and divided into four classes: Normal, CNV, DME, and Drusen. The data is also divided into training and testing sets.

In addition to using the original database, we have also performed data augmentation. This means we have artificially generated new data based on the original ones. The concept and survey of image data augmentation were described by Shorten and Khoshgoftaar [30]. In our research, we used only basic image manipulations such as:Mirror image. Symmetrical reflection of the image in relation to the vertical description of the symmetry of the image. Reflection in relation to the horizontal axis would cause the layers to be inverted, hence it was not used.Rotation. Rotation of the image relative to the center of symmetry of the image. Rotation was carried out in the range of −15°(counterclockwise) to +15°(clockwise) with an interval of 5°.Aspect ratio change. Expanding the image in the range from 105 to 130 percent taking into account the horizontal and vertical axis of the photo separately. Changing both axes simultaneously would only change the image size.Histogram equalization. Equalizing the pixel value histogram. The dependence of the image acquisition on different tissue permeability is reduced.Gaussian blur. Blur with the kernel parameter (5, 5). The operation is to increase the number of samples and add distorted samples-less sharp-based on the original.Sharpen filter. Edge sharpening operation, inverse to blurring, according to:     [[-1, -1, -1],     [-1,  9, -1],     [-1, -1, -1]]

These operations were used to create the augmentation schema presented in Figure 2. The schema uses all images produced in the previous steps in order to multiply the data to a high number of extra data. The arrows on the schema indicate processing workflow.

Using the augmentation schema, we have produced a new image database with a significantly larger number of samples. The number of original samples for training and validation was multiplied 48 times. The original and augmented training set are compared in Table 1. For validation, we used 348 samples for each category in the augmented datasets. Test samples remain unchanged with 242 for each category. In addition, by changing the class distribution of this dataset, a balanced training dataset consisting of 7000 images was created for each category.

## 4. Proposed Models

Inspired by MiniVGGNet, presented in [31] which uses the main architecture of the VGG network [32], we created CNN models with different convolution layers (3, 5, 7, 9, 11) (as shown in Table 2).

Afterward, the proposed lightweight model, with only five convolution layers, was modified to apply to our new augmented dataset.The 5-layer (v1) and 5-layer (v2) network architecture diagram is shown in Figure 3 and Table 3 respectively.

The model parameters and hyperparameters (in two different versions (v1 and v2)) are characterized as follows:Image size = 128 × 128 × 3Batch size = 32Epochs = 10Kernel size (v1) = 3 × 3Kernel size (v2) = 3 × 3 & 5 × 5Max pooling = 3 × 3 & 2 × 2Activation function = ReLU (Rectified Linear Unit)Dropout (v1): 25% & 50%Dropout (v2): 15% & 15% & 15% & 25% & 10% & 10% & 10%Adam optimizer = 0.001 (v1), 0.002 (v2)Loss function = Categorical Cross-EntropyDense (v1): 1024Dense (v2): 256 & 128 & 64 & 32Output layer activation function = SoftmaxNumber of training images = 28,000Number of validation images = 6488Number of test images = 968

In CNN models with many convolution layers (such as the 9-layer, 11-layer, and 5-layer (v2) model in this study) we used separable convolutions instead of basic (standard) convolutions to train the network faster. Our model training process in each epoch was reduced by 0.89 (approximately 2425 s per epoch) using a separable convolution method. Spatially separable convolutions and depthwise separable convolutions are two types of separable convolution methods that aim to split a kernel (N × N) into two smaller kernels (N × 1) and (1 × N) (see Figure 4).

In contrast to the traditional way of convolving an image by using an (N × N) kernel, which requires N × N multiplication, in the separable convolution method, two convolutions with 1 × N and N × 1 kernels with 2 × N multiplication are applied. Spatial separable convolutions can-not divide all kernel types into two separate and smaller kernels. Therefore, depthwise convolution is used, which can be applied to such kernels [33,34,35]. It first performs a depth-wise convolution and afterward applies a 1 × 1 filter to change the dimension (see Figure 5).

## 5. Experiments and Evaluation

To reasonably evaluate model performance in an OCT image dataset with an imbalanced class distribution, several alternative metrics need to be considered [36,37]. In this study, the recall, the precision, the F1- score, G-measure, and accuracy are utilized to evaluate the performance of the proposed model by the following formulae [38,39]:(1)$$\mathrm{Recall}=\frac{TP}{TP+FN}$$
(2)$$\mathrm{Precision}=\frac{TP}{TP+FP}$$
(3)$$\mathrm{F1-score}=\frac{2*recall*precision}{recall+precision}$$
(4)$$\mathrm{G-measure}=\sqrt{precision*recall}$$
(5)$$\mathrm{Accuracy}=\frac{TP}{TP+FN+FP+TN}$$
where TP, FN, and FP, denote the number of true positives, false negatives, and false positives respectively. False-negative (FN) or type 2 error is the most crucial error, in medical image analysis. In the following, we will show that our proposed strategy and the model can classify OCT images with minimal error.

For all patients with eye disease, recall shows the number of correctly identified patients with the same type of eye disease. The precision metric is the measure of patients we correctly identified with specific eye diseases from all patients with the same disease. In medical images, recall is a more important metric than precision, as it is crucial to detect and count patients with actual eye disease. However, both measurements need to be evaluated as they may be indicative of another ailment. Therefore, the F1-score may be a better measure to seek a balance between precision and recall. The main formula for calculating the F-value is:(6)$$F_{\beta }=\left(1+\beta^{2}\right)*\frac{recall*precision}{(\beta^{2}*recall)+precision},\;\beta \geq 0$$
where the value of β is 1, it is known as the F1-score. Contrary to the F-measure, which is calculated using a weighted harmonic mean between precision and sensitivity, the G-measure is the geometric mean of sensitivity and precision. Accuracy (the ratio of correctly predicted observations to total observations) is one of the most intuitive performance measures to use on symmetric datasets [38,39].

In the following, we demonstrate the performance of the proposed algorithms using the aforementioned metrics. Table 4, Table 5, Table 6, Table 7 and Table 8, represent the classification accuracy per class of the proposed CNN models with various convolution layers. A confusion matrix is used to summarize the diagnostic accuracy of each model separately (Figure 6, Figure 7, Figure 8, Figure 9 and Figure 10).

As shown in Table 9, increasing the number of convolution layers does not necessarily improve model accuracy. Adding extra hidden layers to the CNN architecture increases the number of parameters in the network to extract more features from which we can expect better accuracy. Furthermore improving the model performance depends on many other factors such as the number of training and validation datasets, size and resolution of features, etc. The size of the features in our dataset varies. Using a CNN model with fewer hidden layers causes high-level features to go undetected. On the contrary, if the CNN model uses more convolution layers, it may lose low-level features. Furthermore, choosing the right CNN architecture is not just a choice based on the performance of the model, but it is about finding the right trade-off between the accuracy and speed of the model. Therefore in this study data augmentation is applied to the 5-layer model. As seen in Table 10, the application of augmented data to the 5-layer model significantly improved the model accuracy. A confusion matrix was also used to represent the performance of the proposed model on the test dataset (Figure 11).

Using these CNN models trained on the original dataset, we encountered some misclassified images with higher classification confidence scores. Comparing the model classification confidence scores with the image quality results shows that the proposed model enables OCT images to be categorized with the highest confidence score and that the main reason for misclassifications with higher confidence scores is image distortions.

Therefore, using high-quality images or training the model with a large number of augmented images should improve model accuracy.

Accordingly, we implemented two strategies to reduce the effect of noise in images. First, (5 × 5) and (7 × 7) Gaussian kernels with a default border type were used to smooth the image noise. The formula for a Gaussian function in two dimensions is [40,41]:(7)$$G\left(x,y\right)=\frac{1}{2\pi \sigma^{2}}e^{-\frac{x^{2}+y^{2}}{2\sigma^{2}}}$$
where σ is the standard deviation of the Gaussian distribution, *x* and *y* are the distances from the origin in the horizontal and vertical axis respectively. There was not a considerable improvement in model accuracy due to very poor quality OCT images.

The second scenario was to prepare an augmented dataset as described in Section 3. Training the model with a new augmented dataset significantly improved the classification accuracy of the proposed convolutional neural network. The proposed algorithm is compared with 6 state-of-the-art approaches [29] (Table 11). Furthermore, the performance of the proposed algorithm is compared with the transfer learning-based algorithms presented in [24] (Table 12). Models were compared without any changes in their parameters and hyperparameters.

Gradient-weighted Class Activation Mapping (Grad-CAM) (see Figure 12) was also performed to visually confirm the performance of our CNN model in the most significant regions in the OCT image [35,42,43,44].

We also investigated the impact of the class imbalance dataset on the classification performance of 5-layer CNNs (v1 and v2). As can be seen from (Table 13), the performance of the CNN model (v2) improved using the class balanced training dataset, whilst the performance decreased slightly using the v1 model.

Figure 13, Figure 14, Figure 15 and Figure 16 show several examples of the proposed model performance (accuracy and classification confidence score) on OCT images using the original and augmented dataset-trained model, respectively.

It is worth mentioning about obtaining low computational complexity for the prediction using the proposed model. We compared our model with [29]. For the state-of-the-art model, we obtained an average prediction time on the test set of 0.0545 s. For our model, the average time was 0.0471 s, which is an acceleration of about 16.5%. The performance was measured on a Core i7-4771 PC with Nvidia GeForce GTX 760 GPU and 16 GB of RAM. We were using Windows 10 OS, Python 3.8.5, and Keras library 2.4.3.

During the implementation of the solution, we faced several technical difficulties. Due to the size of the set after augmentation, its transfer and use for training were difficult. Training time also increased significantly. In terms of prediction, we did not notice a significant time difference between the model trained without augmentation. During implementation, pay attention to the correct launch of the computing environment for machine learning in order to take advantage of the graphics processing unit (GPU).

## 6. Discussion and Significance of Proposed Work

OCT is a non-invasive imaging method for many diseases. Ophthalmologists agree on the importance of its performance in the process of medical diagnostics. The popularity of the test is constantly growing, which leads us to the rising amount of data to process. An automatic support system for medical staff can improve the quality of their services. The algorithm can work 24 h a day without fatigue. On the other hand, because of the importance of the matter of analysis, it always needs a human operator. The algorithm itself cannot take responsibility for the decision. Therefore, we can thread it as a support system.

The solutions we propose are developed as part of a scientific project whose direct goal is to implement the results. Automation of the OCT image analysis process with the indication of possible pathologies can be valuable information for accelerating diagnostics. The applied nature of the research emphasizes the implementation aspects related to the low demand for computing power. As a result, we propose a simpler model network. An additional goal of introducing the solution to the market is to enable screening in places where an ophthalmologist is not always available (e.g., in opticians’ salons) which is a further plan for the project.

We plan to expand the system by adding additional diseases. To do this, we need to collect a large enough set of labeled data, which is a challenge. We also plan to use information from a set of scans from one examination to make a final decision.

## 7. Conclusions

In the paper, a very lightweight Convolutional Neural Network is proposed for the classification of retinal diseases from Optical Coherence Tomography images. At first, various CNN models with different convolution layers were designed and evaluated. Increasing the number of hidden layers clearly did not improve the final accuracy of the model and led to overfitting. The images used in this study suffer from various degrees of distortion, such as noise, reversing, contrast change, and perspective. We show that the application of augmented images to a suitable convolutional neural network can improve the accuracy outcome in classifying images with greater distortion without the need to increase model complexity.

The proposed CNN model improves the accuracy and speed of state-of-the-art methods. Out of 968 test samples, it achieves 99.90% accuracy with only one misclassification, whereas the best from state-of-the-art [29] is 99.69% with three misclassifications. The proposed method also reduces prediction time by 16.5%, which is important for commercial applications.

As a result of the classification accuracy and speed obtained for the classification of OCT images, the proposed algorithm can be used by experts in medical centers for real-time medical applications. It will be our future research to extend this approach to not only classify OCT images into 4 categories but also to detect the disease’s features within images. Many other eye diseases can be investigated (e.g., central serous chorioretinopathy (CSR), epiretinal membrane (ERM)). Another future research is to extend the algorithm to detect significantly more types of eye diseases using the upgraded model.

## Figures and Tables

**Figure 1 sensors-22-04675-f001:**
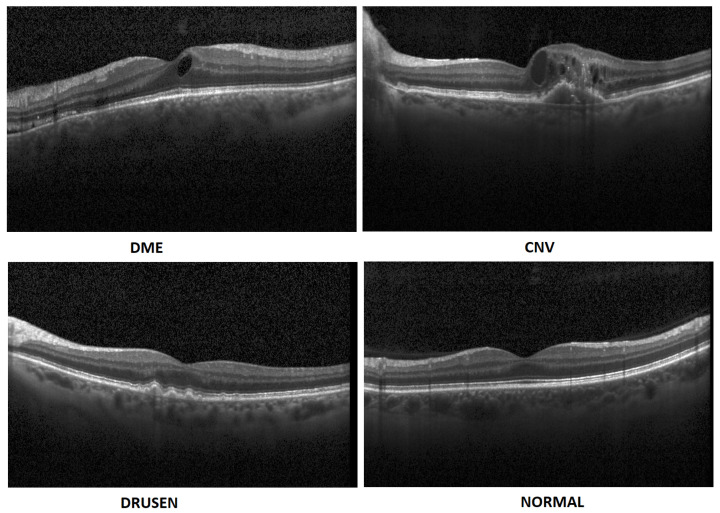
Exemplary OCT B-Scans with diseases and normal image [7].

**Figure 2 sensors-22-04675-f002:**
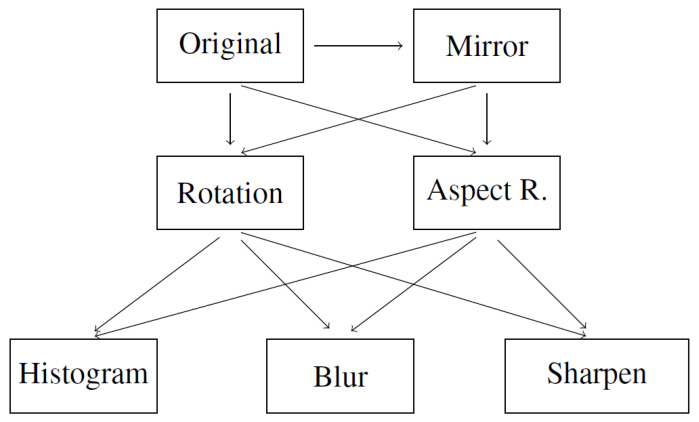
Augmentation schema for artificial data.

**Figure 3 sensors-22-04675-f003:**
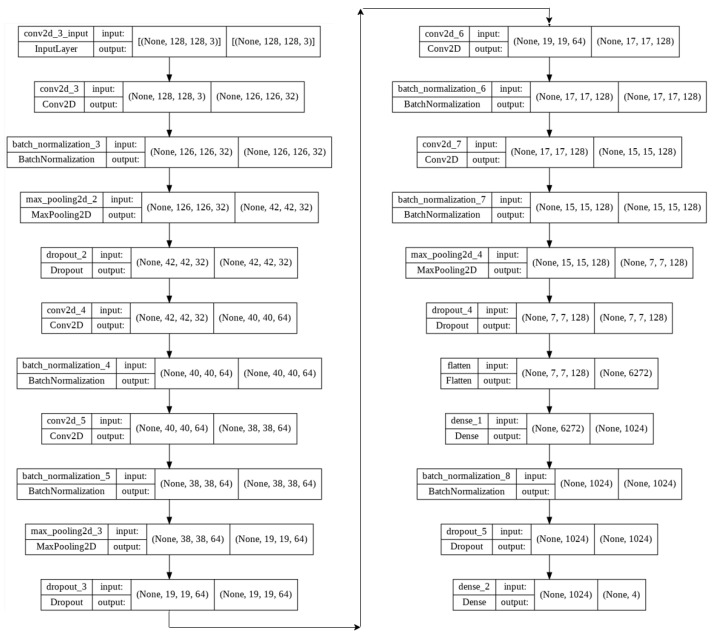
5-layer(v1) network architecture diagram.

**Figure 4 sensors-22-04675-f004:**
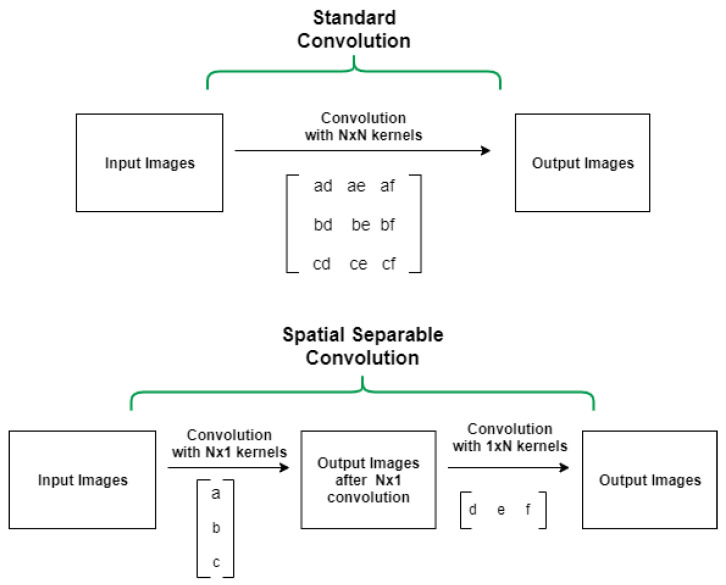
Standard convolution versus spatial separable convolution.

**Figure 5 sensors-22-04675-f005:**
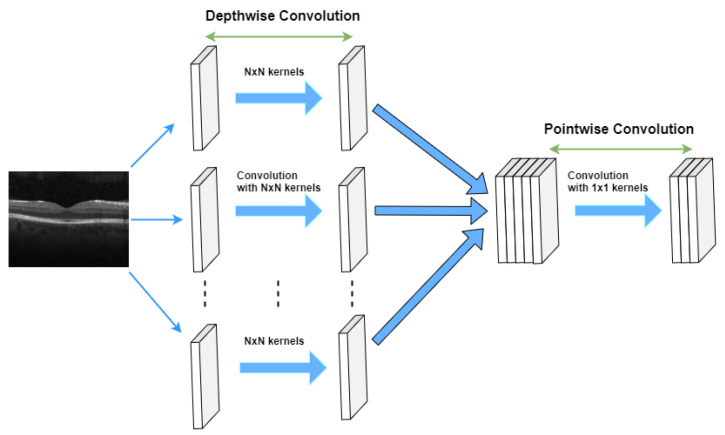
Depthwise separable convolutions.

**Figure 6 sensors-22-04675-f006:**
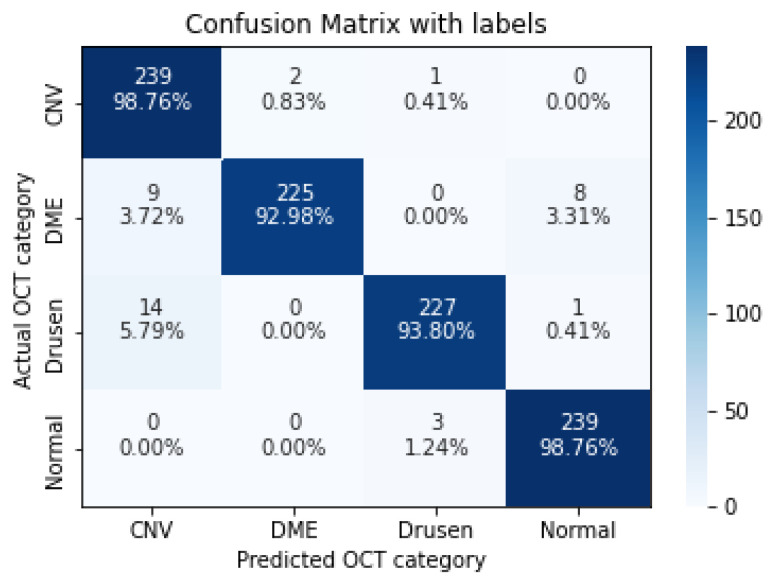
The confusion matrix of 3 layers model, with numbers and percentages.

**Figure 7 sensors-22-04675-f007:**
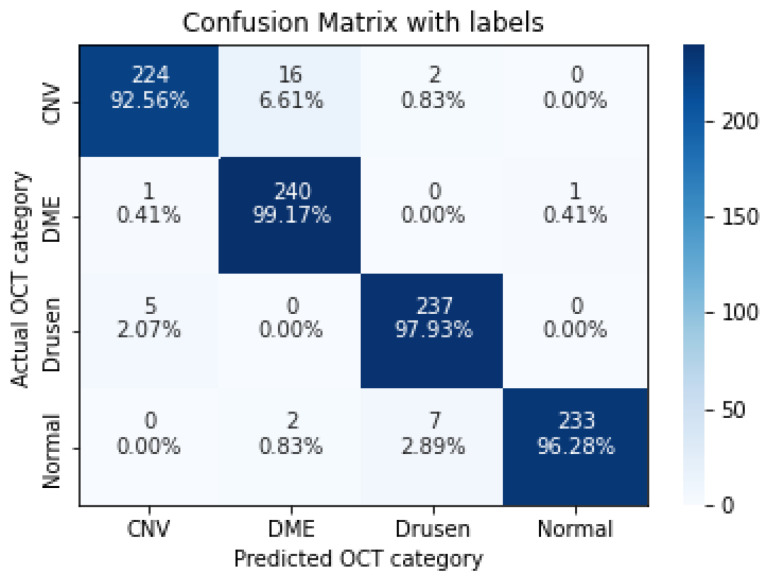
The confusion matrix of 5 layers model, with numbers and percentages.

**Figure 8 sensors-22-04675-f008:**
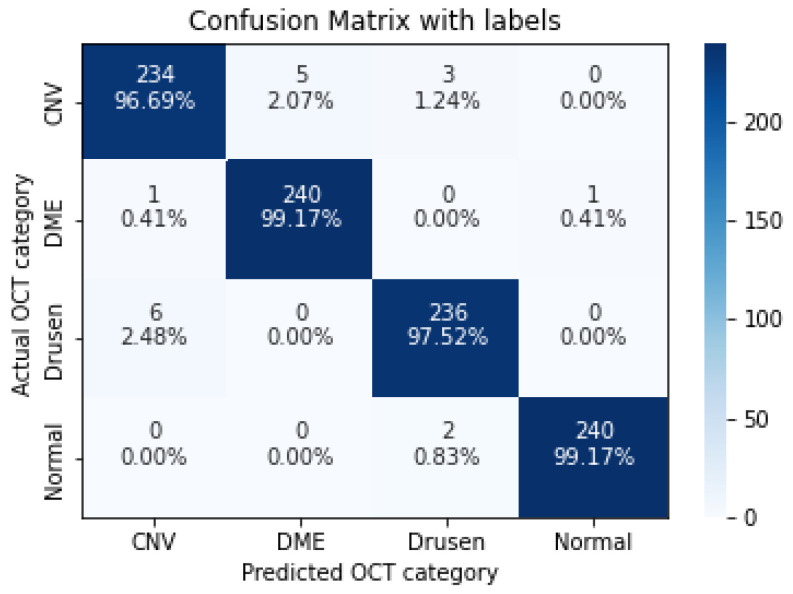
The confusion matrix of 7 layers model, with numbers and percentages.

**Figure 9 sensors-22-04675-f009:**
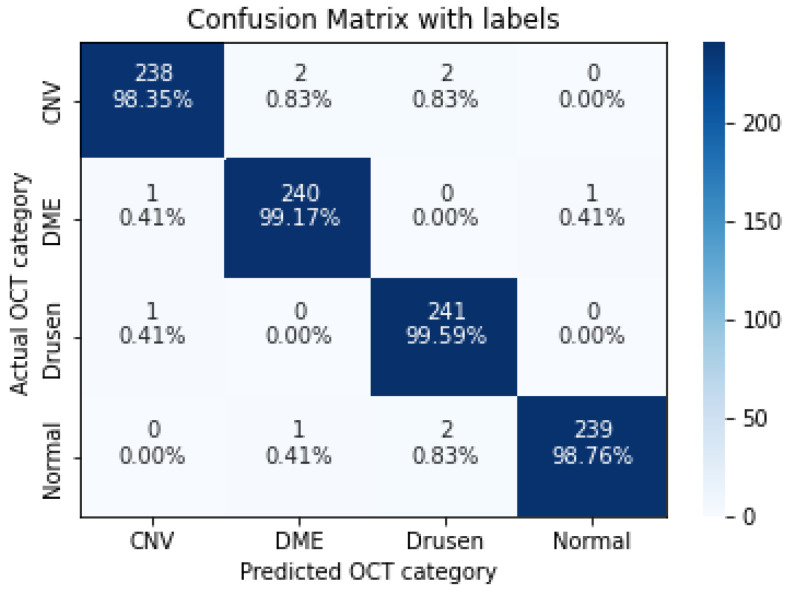
The confusion matrix of 9 layers model, with numbers and percentages.

**Figure 10 sensors-22-04675-f010:**
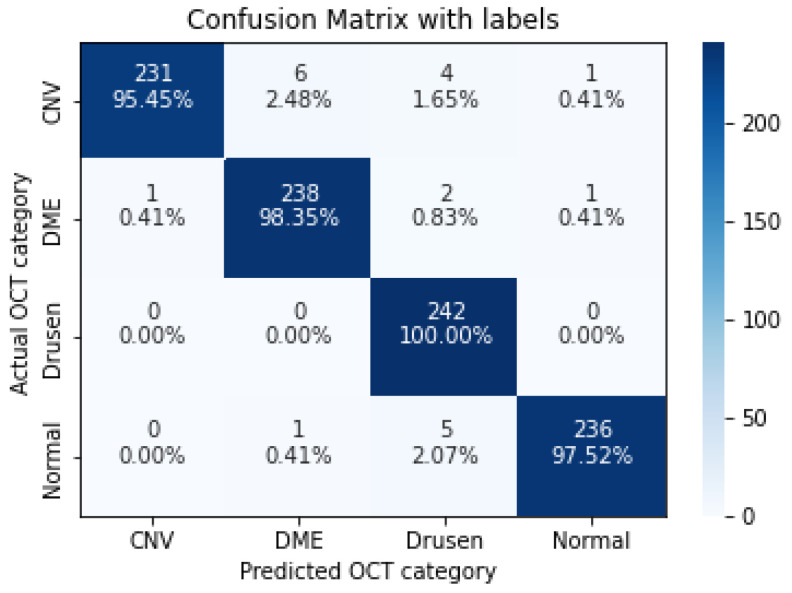
The confusion matrix of 11 layers model, with numbers and percentages.

**Figure 11 sensors-22-04675-f011:**
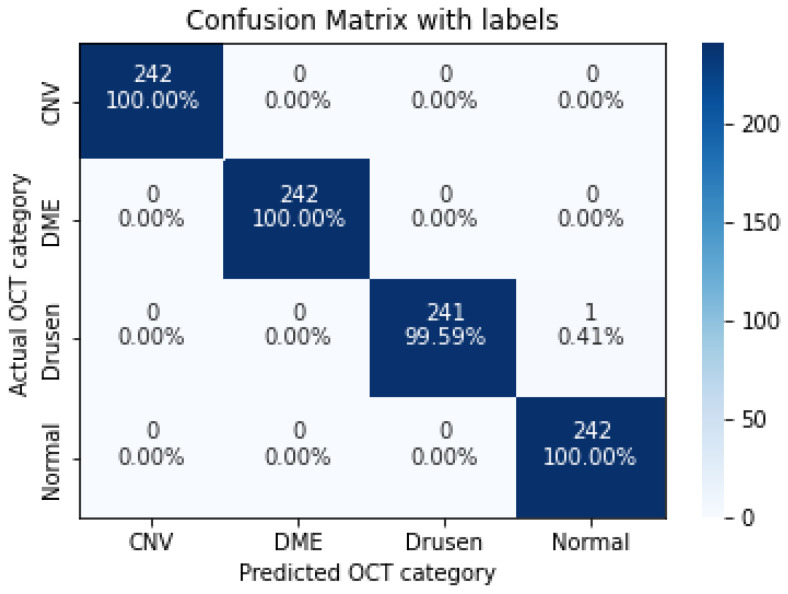
The confusion matrix of 5 layers model (trained over augmented dataset), with numbers and percentages.

**Figure 12 sensors-22-04675-f012:**
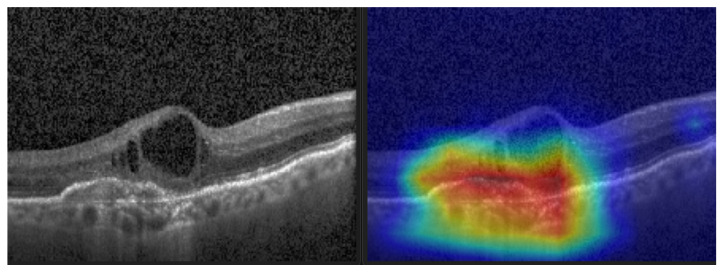
CNV and its Grad-CAM.

**Figure 13 sensors-22-04675-f013:**
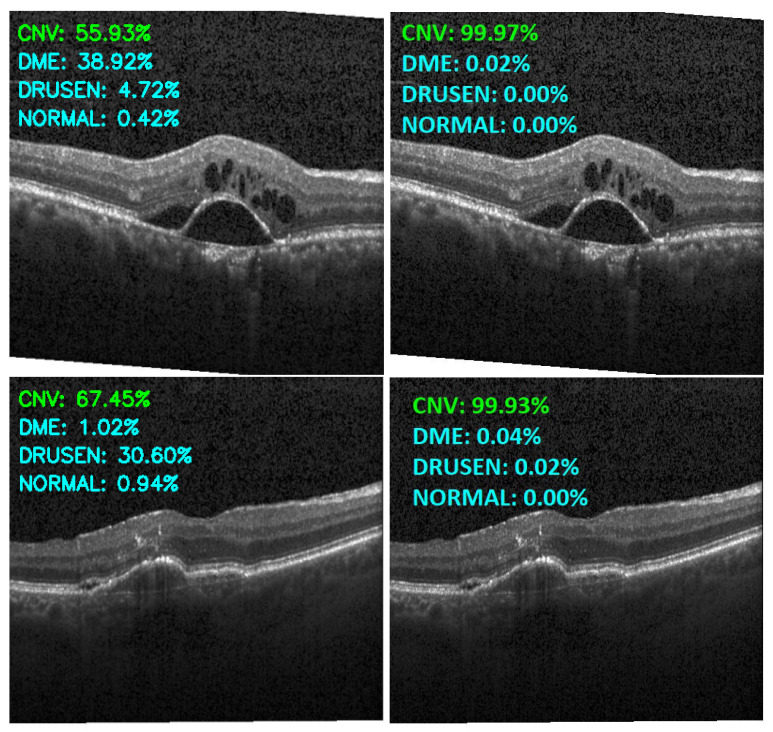
Exemplary result of proposed model trained without (left column) and with (right column) augmentation (correct label highlighted in green).

**Figure 14 sensors-22-04675-f014:**
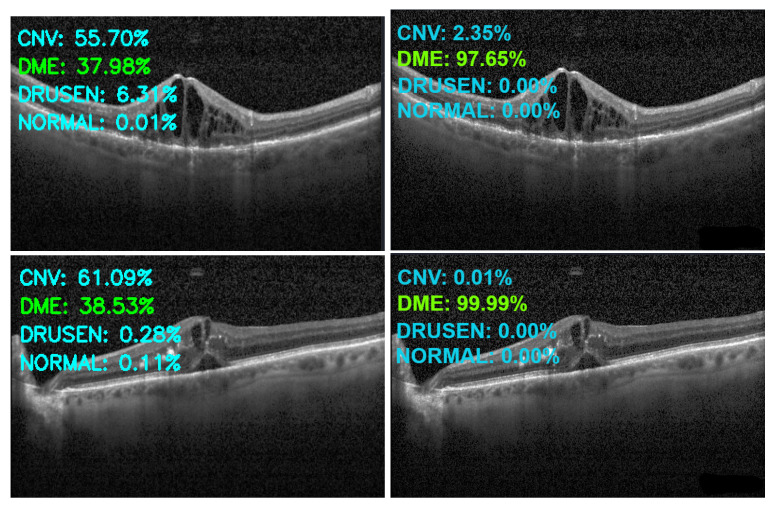
Exemplary result of proposed model trained without (left column) and with (right column) augmentation (correct label highlighted in green).

**Figure 15 sensors-22-04675-f015:**
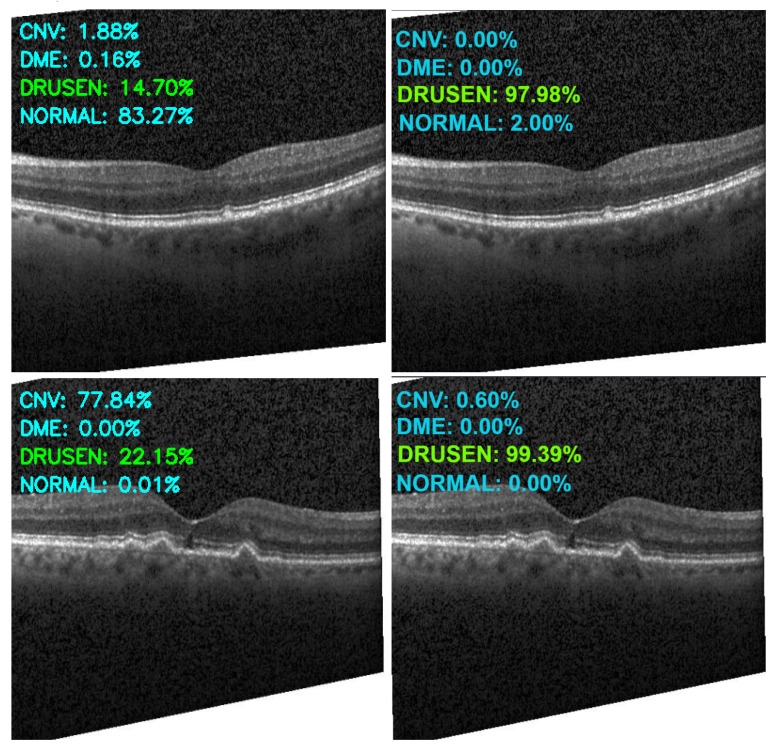
Exemplary result of proposed model trained without (left column) and with (right column) augmentation (correct label highlighted in green).

**Figure 16 sensors-22-04675-f016:**
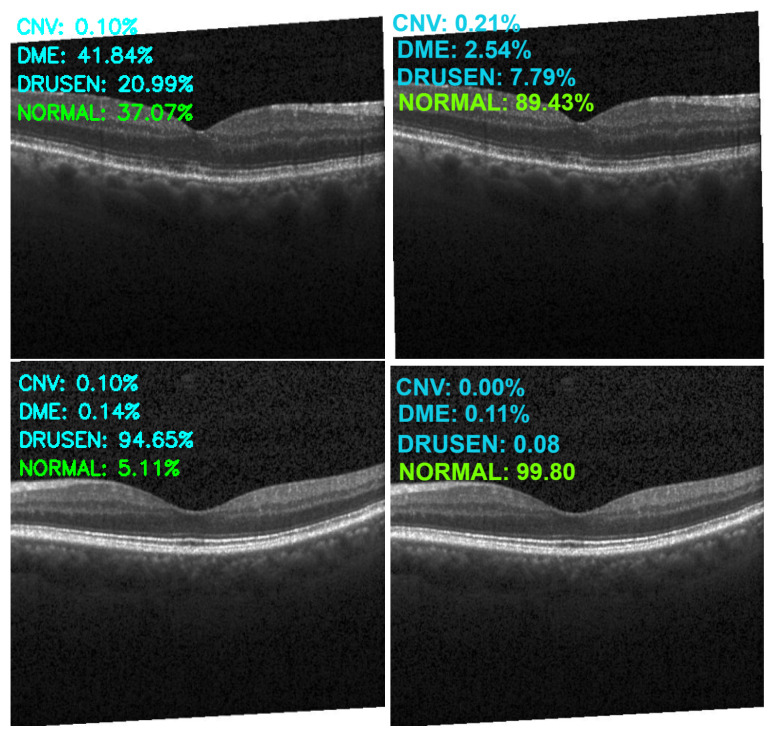
Exemplary result of proposed model trained without (left column) and with (right column) augmentation (correct label highlighted in green).

**Table 1 sensors-22-04675-t001:** Number of training samples for original and augmented dataset.

Category	Original	Augmented
CNV	37,205	1,785,840
DME	11,348	544,704
Drusen	8616	413,568
Normal	26,315	1,263,120

**Table 2 sensors-22-04675-t002:** Proposed CNN model with various number of convolution layers and filters (f).

Model	f1	f2	f3	f4	f5	f6	f7	f8	f9	f10	f11
3 Layers	32	64	128								
5 Layers	32	64	128	256	512						
7 Layers	32	32	64	64	128	256	512				
9 Layers	32	32	64	64	128	128	128	512	512		
11 Layers	32	32	64	64	128	128	256	256	512	512	512

**Table 3 sensors-22-04675-t003:** 5-Layers (v2) network architecture.

Layer (Type)	Output Shape	Param #
Separable Conv2D	(None, 126, 126, 128)	539
Batch_Normalization	(None, 126, 126, 128)	512
Max_Pooling2D	(None, 42, 42, 128)	0
Dropout	(None, 42, 42, 128)	0
Separable Conv2D	(None, 40, 40, 128)	17,664
Batch_Normalization	(None, 40, 40, 128)	512
Separable Conv2D	(None, 36, 36, 128)	19,712
Batch_Normalization	(None, 36, 36, 128)	512
Max_Pooling2D	(None, 18, 18, 128)	0
Dropout	(None, 18, 18, 128)	0
Separable Conv2D	(None, 16, 16, 256)	34,176
Batch_Normalization	(None, 16, 16, 256)	1024
Separable Conv2D	(None, 12, 12, 256)	72,192
Batch_Normalization	(None, 12, 12, 256)	1024
Max_Pooling2D	(None, 6, 6, 256)	0
Dropout	(None, 6, 6, 256)	0
Flatten	(None, 9216)	0
Dense	(None, 256)	2,359,552
Batch_Normalization	(None, 256)	1024
Dropout	(None, 256)	0
Dense	(None, 128)	32,896
Batch_Normalization	(None, 128)	512
Dropout	(None, 128)	0
Dense	(None, 64)	8256
Batch_Normalization	(None, 64)	256
Dropout	(None, 64)	0
Dense	(None, 32)	2080
Batch_Normalization	(None, 32)	128
Dropout	(None, 32)	0
Dense	(None, 4)	132
Total Pramas:		2,552,703
Trainable Pramas:		2,549,951
Non-Trainable Pramas		2752

**Table 4 sensors-22-04675-t004:** CNN model with 3 hidden layers.

	Recall	Precision	F1-Score
CNV	0.9876	0.9122	0.9484
DME	0.9298	0.9912	0.9595
Drusen	0.9380	0.9827	0.9598
Normal	0.9876	0.9637	0.9755

**Table 5 sensors-22-04675-t005:** CNN model with 5 hidden layers.

	Recall	Precision	F1-Score
CNV	0.9256	0.9739	0.9492
DME	0.9917	0.9302	0.9600
Drusen	0.9793	0.9634	0.9713
Normal	0.9628	0.9957	0.9790

**Table 6 sensors-22-04675-t006:** CNN model with 7 hidden layers.

	Recall	Precision	F1-Score
CNV	0.9669	0.9710	0.9689
DME	0.9917	0.9796	0.9856
Drusen	0.9752	0.9793	0.9772
Normal	0.9917	0.9959	0.9938

**Table 7 sensors-22-04675-t007:** CNN model with 9 hidden hayers.

	Recall	Precision	F1-Score
CNV	0.9835	0.9917	0.9876
DME	0.9917	0.9877	0.9897
Drusen	0.9959	0.9837	0.9897
Normal	0.9876	0.9958	0.9917

**Table 8 sensors-22-04675-t008:** CNN model with 11 hidden layers.

	Recall	Precision	F1-Score
CNV	0.9545	0.9957	0.9747
DME	0.9835	0.9714	0.9774
Drusen	1.0000	0.9565	0.9778
Normal	0.9752	0.9916	0.9833

**Table 9 sensors-22-04675-t009:** Comparison of CNN models with various convolution layers.

Number of Convolutions	Recall	Precision	F1-Score	G-Measure
3-Layers	0.9607	0.9624	0.9608	0.9615
5-Layers	0.9649	0.9658	0.9649	0.9653
7-Layers	0.9814	0.9814	0.9814	0.9814
9-Layers	0.9897	0.9897	0.9897	0.9897
11-Layers	0.9783	0.9788	0.9783	0.9785

**Table 10 sensors-22-04675-t010:** Proposed CNN model with 5 hidden layers trained over augmented data.

	Recall	Precision	F1-Score
CNV	1.0000	1.0000	1.0000
DME	1.0000	1.0000	1.0000
Drusen	0.9959	1.0000	0.9979
Normal	1.0000	0.9959	0.9979

**Table 11 sensors-22-04675-t011:** Comparison of methods.

Algorithms	Recall	Precision	F1-Score	Accuracy
Proposed Method	0.9990	0.9990	0.9990	0.9990
A.P. Sunija et al. [29]	0.9969	0.9969	0.9968	0.9969
D.S. Kermany et al. [7]	0.9780	N/A	N/A	0.9660
L. Huang et al. [19]	N/A	N/A	N/A	0.8990
S. Kaymak et al. [25]	0.9960	N/A	N/A	0.9710
V. Das et al. [22]	0.9960	0.9960	0.9960	0.9960
D.K. Hwang. [23]	N/A	N/A	N/A	0.9693

**Table 12 sensors-22-04675-t012:** Comparative analysis of the proposed method with the methods presented in [24].

Algorithms	Resolution	Recall	Precision	F1-Score	Accuracy
Proposed Method	128 × 128 × 3	0.9990	0.9990	0.9990	0.9990
CNN	150 × 150 × 3	0.98	0.98	0.98	0.98
Xception	150 × 150 × 3	0.99	0.99	0.99	0.99
ResNet-50	150 × 150 × 3	0.97	0.97	0.97	0.97
MobileNet-V2	150 × 150 × 3	0.99	0.99	0.99	0.99

**Table 13 sensors-22-04675-t013:** Performance of 5-layer CNN models trained on the original dataset; (Comparison based on Class Balanced and Imbalanced Dataset).

CNN Model	Dataset Distribution	Recall	Precision	F1-Score
5-Layers-v1	(imbalanced)	0.9804	0.9806	0.9804
5-Layers-v1	(balanced)	0.9752	0.9754	0.9752
5-Layers-v2	(imbalanced)	0.9783	0.9794	0.9784
5-Layers-v2	(balanced)	0.9886	0.9887	0.9886

## Data Availability

Research presented in the article was performed using publicly available image dataset for OCT scan available on: https://www.kaggle.com/datasets/paultimothymooney/kermany2018, accessed on 1 May 2022.

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
