# Peer review of "Fast and Efficient Method for Optical Coherence Tomography Images Classification Using Deep Learning Approach"

_sensors, 2022, doi:10.3390/s22134675_

Round 1

Reviewer 1 Report

The paper proposes a new approach of convolutional neural network for OCT images classification of eye diseases. After testing, this model has higher effectiveness and improves computation speed.

The main concerns are listed as follow:

1, What are the two different versions (v1 and v2) mentioned in the fourth part? Are they the original network and the lightweight network?

2, Section 4.1 describes “Spatial separable convolutions can-not split any kind of kernels into two separate and smaller kernels”, but depthwise separable convolutions does not seem to solve this problem and does not split the kernel into two.

3, How can you get the conclusion of choose 5 hidden layers from the confusion matrix tables?

4, You can add the comparison of the results of training with the original dataset and training with the augmented dataset.

5, Please add the ablation experiments as a comparison.

6. Some machine learning works are missed, eg. Deep-IRTarget: An Automatic Target Detector in Infrared Imagery using Dual-domain Feature Extraction and Allocation

Reviewer 2 Report

In this paper, authors presented an approach for the classification of eye diseases using the convolutional neural network model. The research concerns the classification of patients on the basis of OCT B-scans into one of four categories. The following review comments are recommended, and the authors are invited to explain and modify.

Comment: It is best to avoid using abbreviations in the abstract unless the abbreviation is commonly understood and/or is used multiple times in the abstract. Authors should also write in full like OCT, DME, CNV etc.

Comment:  The abstract section is inconsistent and does not reflect the main contributions of the manuscript. The authors should rewrite the abstract section to mention the main purpose of the paper, primary contributions, experimental results, and global implications.

Comment:  “We are proposing in the paper a high accuracy solution with a very light CNN model”; could the authors explain better what they mean and how they did it to support their claim in getting results?

Comment: Novelty is confusing. A highlight is required. The main contributions of the manuscript are not clear. The main contributions of the ‎article must be very clear and would be better if summarize ‎them into 3-4 points at the ‎end of the introduction.‎

Comment: “The model parameters and hyperparameters (in two different versions (v1 and v2)) are characterized as follows”, how to optimize hyper parameters?

Comment: Nothing is mentioned about the implementation challenges.

Comment: The following clinical decision support systems using Deep Learning, and medical imaging must be included to improve the quality of the paper.

·  A Lightweight Convolutional Neural Network Model for Liver Segmentation in Medical Diagnosis

·  OP-convNet: A Patch Classification-Based Framework for CT Vertebrae Segmentation

Comment: Discuss the stability of the system in terms of complexity.

Comment: Moreover, it should be noticed that the clinical appliance has to be decided by medicals since the existing differences between the real image and the one generated by the proposed system could be substantial in the medical field.

Comment: Could you please check your references carefully (in particular, proceedings: location of the conference, date of the conference, publisher's name and location...)? All references must be complete before the acceptance of a manuscript.

Reviewer 3 Report

Reviewer’s Report on the manuscript entitled:

Fast and Efficient Method for OCT Images Classification using Deep Learning Approach

The authors proposed a new approach for the classification of eye diseases using the convolutional neural network model. The methods and results are interesting; however, the presentation needs to be improved. Please see below my comments.

Please write the full name for OCT in the title of the manuscript and remove OCT.

Please define the abbreviations OCT, DME, and CNV in the Abstract.

Please define the abbreviations the first time they appear and be consistent with their style/format.

Please remove lines 7, 8,9,10 from the Abstract. Simply say something like, “Our results showed higher accuracy and computational efficiency compared to the results of other similar methods.”

Line 26. Please add the following most recent articles showing the use of various artificial intelligence methods in medicine

https://doi.org/10.3390/s22062346

https://doi.org/10.3390/app9132700

https://doi.org/10.3390/s22082948

Line 26. Please replace “In article [3], the authors…” with “Garcia and Simunik [3] …”. In other words, please give the authors' names. Please check and correct elsewhere.  

Line 86. By referring to the Section number, please describe how the rest of the manuscript is organized.

Figure 3 must be improved. There is plenty of blank space on each side of the flowchart to move the boxes around and enlarge the font size of the texts.

Line 149. Please remove the heading.

Table 12 and lines 217-222. Are all these methods applied to the same datasets? If so, why Recall, Precision and F1-score are reported as N/A for some of the methods? Please explain.

Figure 14. Please enlarge the font size of the texts in the right panels to have the same size as the left panels. I see that Figures 13,14,15,16 have four panels each a separate figure. Please generate one figure with four panels in it instead and make the font size consistent for all of them.

The Discussion part (Section 6) is very short. Please add a few more paragraphs to discuss the results, advantages, and disadvantages of one method with respect to another and provide recommendations and areas of improvement.

Thank you for your contribution

Regards,

Round 2

Reviewer 1 Report

accept

Author Response

Thank you for accepting our article. We have made additional language and spell check.

Reviewer 2 Report

The authors answered my questions satisfactorily.

Author Response

(The authors gave the same response as above.)

Reviewer 3 Report

I would like to thank the authors for addressing my comments. The manuscript is improved in my view and I recommend acceptance after some minor revisions.

Line 61. Grammar issue. "Many researchers deal with..." not "...deals with".

Line 68. Remove the second comma.

Line 97. Remove "The" before Lo et al. [27]. Also, it is "et al." not "et. al."

Figure 3. Improve the flowchart. Please move some of the boxes to the left and/or right so you can enlarge the font size.

Line 265. I think it is "were" not "was".

Line 282. "...we propose a simpler..."

Line 284. Please replace "what" with "that"

Line 355. Please write the article number after the volume number. Here for example, you should say "... 2022, 22, 2948". Similarly, insert the article number after the volume numbers for articles [12], [14], etc. Please carefully check the references and follow the MDPI guidelines.

Please very carefully proofread the article before publication if accepted by the editor.

Thank you!

Author Response

Thank you. We have fix mentioned issues. We have checked article ones again. We change the figure 3 into 2 columns style. We have also fix the references (the problem was that Bibtex does not catch ARTICLE-NUMBER fields properly.